# Postharvest Dips of Persimmon Fruit in Gibberellic Acid: An Efficient Treatment to Improve Storability and Reduce Alternaria Black Spot

**Dalia Maurer [1], Oleg Feygenberg [1], Alon Tzoor [2], Guy Atzmon [2], Shlomo Glidai [2] and Dov Prusky [1,\*]**

[1] Department of Postharvest Science of Fresh Produce, The Volcani Center, POB 6, Bet Dagan 50250, Israel; daliam@volcani.agri.gov.il (D.M.); fgboleg@volcani.agri.gov.il (O.F.)

[2] Gadot Agro, Gadot Group, Kidron 7079500, Israel; alonz@gadot.com (A.T.); guya@gadot.com (G.A.); shlomog@gadot.com (S.G.)

\* Correspondence: dovprusk@agri.gov.il

**Abstract:** In Israel, fruit softening during storage and the occurrence of Alternaria black spot (ABS) disease, caused by *Alternaria alternata*, are the main postharvest factors that reduce quality and impair storability of persimmon fruit. The pathogen causing ABS infects the fruit in the orchard and remains quiescent until harvest, or renews its development just before harvest but mainly during storage. A preharvest spray with 50 $\mu g \cdot L^{-1}$ gibberellin (GA$_3$) significantly improved fruit storability, as determined by fruit firmness and levels of ABS. While GA$_3$ treatments have been commercially applied for more than 30 years, significant limitations of the use of the preharvest treatment like enhancing the risk of a reduced yield have been described. Recent findings suggested that postharvest dip treatments with increased concentrations of GA$_3$ also delayed fruit softening and reduced ABS to similar levels to the commercially applied preharvest treatment in persimmon fruit stored for 3 months at 0 °C. Postharvest GA$_3$ dip treatments at concentrations ranging from 500 to 1500 $\mu g \cdot L^{-1}$ were similarly more efficient in the prevention of fruit softening and ABS development than the 50 $\mu g \cdot L^{-1}$ preharvest spray. Present results indicated that postharvest GA$_3$ treatment physiologically affects fruit firmness and susceptibility to ABS during storage.

**Keywords:** improve storage; postharvest diseases; Alternaria; quiescent infections

## 1. Introduction

Harvesting of persimmon fruit is selective and is carried out according to their external color. At a late stage of maturation, i.e., about two weeks prior to harvest, the fruit color shifts from green to slightly orange, and at this time the fruit are treated with gibberellin (GA$_3$) at 50 $\mu g \cdot L^{-1}$, to prevent softening [1–3], and to reduce Alternaria black spot disease (ABS) (*Alternaria alternata*) occurrence during storage [4–6]. Field application of GA$_3$ delayed fruit maturity, postponed the green-to-orange color change, enhanced erection of the sepal lobes and thereby reduced local humidity and ABS severity, and affected the fruit cell wall composition [1,3,4]. The effect of GA$_3$ on cell walls of treated fruit reduced ABS endoglucanase (EG) activity, thereby preventing fungal colonization [1,4].

ABS development is also affected by pre- and postharvest fungicide treatments that protect against and/or eradicate fungal infections, or a combination of both of these approaches during a high incidence of infection [7,8]. After color change and harvest, and prior to storage at 0 °C, fruit are subjected to a dip treatment in a chlorine compound that is released from Troclosen Na tablets [6].

Commercially, a combination of preharvest GA$_3$ treatment and postharvest Troclosen Na dip are effective for the improvement of storability of the fruit at 0 °C for up to three months. However, several

reports indicated that this type of GA$_3$ spray in autumn had a significant effect on the physiology of the tree and contributed to the reduction of yield during the following year [9]. Winer and Bendik [9] reported that this effect could reach up to a 25% and a 26.2% reduction in total yield and percent of exported fruits, respectively. Given that GA$_3$ is a commercially-applied treatment in persimmon in Israel, the objective of the present study was to develop an efficient postharvest treatment to improve storability and the reduction of ABS without the need for a preharvest spray of GA$_3$. The specific objective of the present study was to determine if a postharvest dip in GA$_3$ might enhance fruit quality of persimmon fruit similar to the pre-harvest GA$_3$ spray in the orchard.

## 2. Materials and Methods

### 2.1. Pathogen, Host and Commercial Treatments

Experiments were carried out with persimmon (Diospyros kaki Thunb. cv. Triumph) fruit during two consecutive years (2016–2017), in orchards (Ein Carmel-2016 in the north of Israel, Bar Ness-2017 in the south of Israel) where fruit usually showed high natural incidences of ABS infection during storage. Harvested fruit were stored at 0 °C and about 90% RH for 3 months in 350 kg plastic bins, pending assessment. Symptoms of naturally infected fruits were detected by direct evaluation and the percent colonization was assessed by comparing with established scales of infected fruits covered with Alternaria symptoms [4].

### 2.2. Preharvest Treatments of Persimmon Trees with GA$_3$

The growth regulator GA$_3$ (Giberllon containing gibberellic acid technical at 50 g.L$^{-1}$, Fine Agrochemicals Ltd., Whittington, Worcester, UK) was applied according to commercial practice in 0.025% (v/v) Triton-X100, at 800 L. ha$^{-1}$, 7–10 days before harvest [1]. The effect of GA$_3$ treatment on persimmon fruit quality was tested in a random split-block design, spread along four rows of trees, with six replicates. Each replicate was comprised of six trees with three trees separating the replicates. Harvested fruit placed in 350 kg bins from each group of trees were dipped in a suspension of Troclosen Na at 50 µg·L$^{-1}$ and transferred to storage in commercial conditions at 0 °C. After 3 months of storage, samples of 80 fruit from each bin (80 fruit x 6 bins= 480 fruit) were transferred from the commercial storage room to the Department of Postharvest Science at the ARO, Bet Dagan, Israel, for evaluation.

### 2.3. Postharvest Treatments of GA$_3$

Untreated fruits at the preharvest stage were harvested, treated with a solution of Troclosen Na for 30 s, and the bin was allowed to rinse and dry for 2–3 h. After the fruit were dried, a second 30 s dip in one of two different GA$_3$ formulations was carried out. The first formulation contained GA$_3$ at 50 µg·L$^{-1}$, similar to the formulation used in the preharvest spray, and the second formulation contained GA$_3$ at a concentration of 100 µg·L$^{-1}$, in an oil dispersion formulation (GibberllonOD (GA$_3$OD), Fine Agrochemicals Ltd., Whittington, Worcester, UK). The GA$_3$ treatments were applied at 500, 750, 1000 and 1500$^{-1}$ in 0.025% (v/v) Triton-X100. After 3 months of storage, samples of 80 fruit from each bin (80 fruit × 6 bins = 480 fruit per treatment) were transferred from the commercial storage room for evaluation.

### 2.4. Comparison of Pre- and Postharvest GA$_3$ Treatments in Commercial Orchards

To demonstrate the efficiency of the postharvest dip treatment at a commercial level, the effect of a GA$_3$ dip at 1000 µg·L$^{-1}$ on the delay of fruit softening and reduction of ABS levels was compared to a preharvest GA$_3$ application in two independent orchards, Ein Carmel (in the north of Israel) and Bar Ness (in the south of Israel) with fruit harvested during the years 2016–2017. Harvested fruit placed in 350 kg bins from GA$_3$ commercially treated trees at 50 µg·L$^{-1}$ and GA$_3$ untreated trees were dipped in a suspension of Troclosen Na. Fruits from GA$_3$ untreated tress we postharvest dip treated at 1000 µg·L$^{-1}$ of GA$_3$. All the fruit was transferred to storage in commercial conditions at 0 °C. After

3 months of storage, samples of 80 fruit from each bin (80 fruit x 6 bins= 480 fruit) were transferred from the commercial storage room to the Department of Postharvest Science at the ARO, Bet Dagan, Israel, for evaluation. Fruit of six replications sampled from 6 different bins of the treatment were evaluated for firmness and for ABS affected area as described earlier.

### 2.5. Persimmon Fruit Firmness Index and ABS Severity after Storage

Firmness was assessed by hand pressure, according to a 10 point firmness scale ranging from 1 (soft) to 10 (firm) with values of 6 for firm turning to flexible, and 4 for fruit turning from flexible to soft. Forty fruit were tested per replicate. Firmness was also evaluated by a hand Force Gauge penetrometer (Tension & Compression) (Lutron, FG-20Kg) measuring the firmness in Newtons, using a conical tip on both sides of 25 fruit of each replicate.

The percent of ABS disease severity was recorded as the percentage of the surface area exhibiting black spot decay. Fruit was assessed following 3 months of storage at 0 °C [10] and a fruit was regarded as unmarketable when more than 1% of its surface area was covered by ABS.

### 2.6. Color Evaluation of Fruits

The skin color (hue) of 10 persimmon fruit was measured from the different treatments using a CR-400/410 Chromometer (Konica Minolta, Osaka, Japan) at four points on the shoulder and bottom part of each fruit (20 measurements per treatment) [11]. The hue angle measured color (120 represents a green color; 60–70 represents a yellow color; 30–40 represents a red color).

### 2.7. Statistical Methods

The experiments were carried out during three consecutive years. The results of the final year's experiment, which included all the combinations, are presented. The experimental results obtained in all three years were statistically consistent, and the same pattern of changes was observed. Data were subjected to analysis of variance (ANOVA) by means of the Tukey–Kramer honestly significant difference (HSD) Test at $P < 0.05$.

## 3. Results and Discussion

### 3.1. Effects of Pre- and Postharvest $GA_3$ Treatments on the Firmness of Persimmon Fruit

Persimmon fruit treated at the postharvest stage with $GA_3$ or $GA_3OD$ showed similar firmness indices as the preharvest treated fruit after 3 months of storage at 0 °C. The preharvest treated fruit showed a firmness index values of 7.3 compared to values of 6.7 of untreated fruit. Fruit treated with 500–1500 $\mu g \cdot L^{-1}$ showed a higher firmness index ranging from values of 7.5 to 8.0 compared to the value of 6.7 for untreated fruit (Figure 1A). No differences were found in the firmness index between the fruit response to different concentrations of the $GA_3$ and $GA_3OD$ formulations applied postharvest. However both showed an increased firmness with the increase in the growth regulator concentration.

When firmness of fruit was evaluated with a penetrometer, the average firmness value at harvest was 32.6 N. Fruit firmness in preharvest treated fruit declined to values of 25 N while the firmness of untreated fruit declined to 16 N (Figure 1B). Postharvest dipped fruit at concentrations of 500 to 1500 $\mu g \cdot L^{-1}$ preserved fruit firmness at values ranging between 17–25 N at a range of 21-27 N when treated with the $GA_3OD$ formulation. Altogether, the results indicated the possibility of preventing fruit softening in persimmons by postharvest treatments with $GA_3$.

The enhanced firmness of the fruits may have resulted from the effect of $GA_3$ on the cell wall of ripening fruit. Ben-Arie et al. [1] reported that $GA_3$ treatment caused a 37% higher cellulose content in the cell walls of the treated fruit. Given that cellulose microfibrils provide the structure and support for all the components of plant cell walls, $GA_3$ effects on microfibrils could explain the increased fruit firmness observed after the 3 months of cold storage of treated persimmons [12,13]. At the same time, Ben-Arie, et al. [1] reported that fruit endoglucanase activity was not detectable in $GA_3$-treated fruit.

Both factors, cellulose synthesis and reduced hydrolytic activity, probably affected fruit firmness in the GA$_3$ treatments. The effect of preharvest GA$_3$ treatment on fruit firmness was also reported in sweet cherries [14] and strawberries [15]. In those cases single spray applications before harvest at 30 µg·L$^{-1}$ to sweet cherries and 50 µg·L$^{-1}$ to strawberries resulted in firmer and larger fruit that were harvested later than untreated fruit.

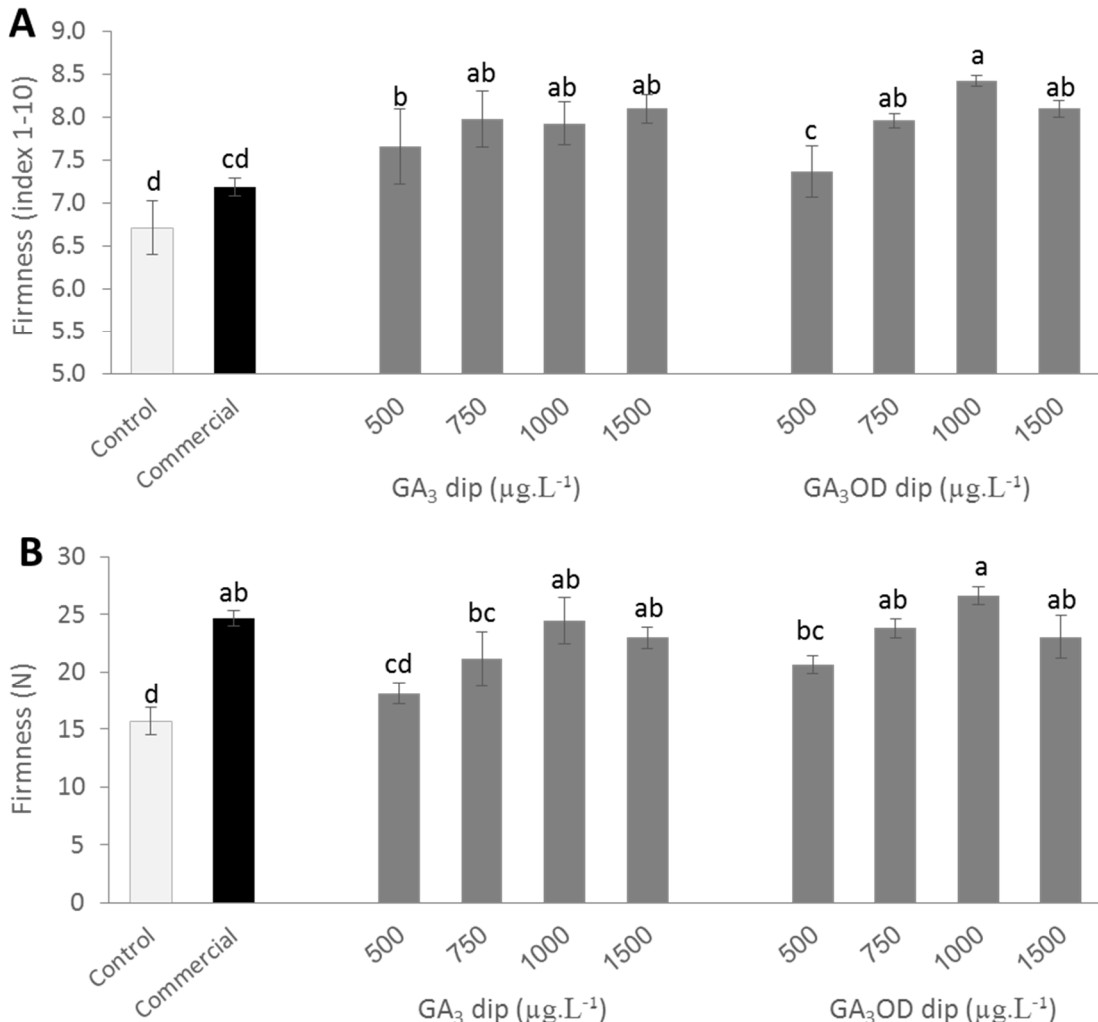

**Figure 1.** Effect of preharvest spray (commercial) and postharvest dip treatments with gibberellin (GA$_3$), using the formulations GA$_3$ and GA$_3$OD, on firmness index (hand pressure: from 1, soft to 10, firm, (**A**)) and firmness (Newton, (**B**)) of persimmon fruit cv. Triumph after 3 months of storage at 0 °C. Average values ($\pm$ standard error) of six replications with different letters are significantly different at $P \leq 0.05$ according to the Tukey–Kramer Multiple Comparison Test.

However, no reports in the past have suggested the possibility of improving persimmon fruit storability by using postharvest treatments with growth regulators. The possibility of shifting from a pre- to postharvest treatment of persimmon would be of extreme importance since it would reduce the need to enter the orchard using agricultural tools, spraying the whole tree, and potentially wounding and breaking branches. Postharvest treatments would improve the homogeneity of the treatment and reduce the effect on yield reduction induced by the autumn GA$_3$ treatment in the following years [9].

The mechanism of regulation of fruit firmness by GA$_3$ is still unclear. However, it was reported that GA$_3$ signaling pathways have been implicated in plant responses to biotic and abiotic stress by modulating salicylic acid (SA) levels, which result from responses to diverse stressful conditions [16]. Lee and Park [16] indicated that GA$_3$ application neutralized the inhibitory effects of salt, oxidative,

and heat stresses during Arabidopsis development accompanied by an increase in SA levels. However, the possibility that $GA_3$ treatments would show active responses at different maturation stages was not expected. The efficiency of $GA_3$ applied on orange (i.e., more mature) persimmon fruit after harvest, indicated a fruit response after harvest. In this case, the postharvest dip treatment was efficient when applied at a more mature physiological stage; however, its efficiency was dependent on a 10–30 fold increase in concentration of the $GA_3$ treatment to 1000 $\mu g \cdot L^{-1}$, compared to 50 $\mu g \cdot L^{-1}$ used as a preharvest treatment.

### 3.2. Effects of Pre and Postharvest Treatment with $GA_3$ on ABS Development of Persimmon Fruit

Comparison of the area covered by ABS developed after 3 months of storage of untreated fruit compared to preharvest-$GA_3$-sprayed-fruit showed a decline in infected area of 33% (Figure 2A). Dip treatment of harvested fruits with $GA_3$ at concentrations from 500 to 1000 $\mu g \cdot L^{-1}$ showed a further decline in the infected area by 42 to 83%, respectively. This response indicated that the postharvest treatment with $GA_3$ was very efficient in inhibition of fungal development, suggesting that the effect of the postharvest $GA_3$ treatment was not only through the effect of $GA_3$ on the host firmness response, but also through the activation of a mechanism of host resistance [4]. In this aspect, $GA_3$ signaling pathways have been implicated in plant response to biotic and abiotic stress by modulating SA [16]. SA is an important defense hormone mediating signal transduction systems, which can stimulate both localized acquired resistance and systemic acquired resistance. SA is predominantly correlated with resistance against biotrophic and hemibiotrophic pathogens. This may explain the effect of $GA_3$ in the induced resistance of persimmon fruit while under the quiescent state of Alternaria, where the hyphae is still non-active [15]. This may indicate that the growth regulator may reduce the development of ABS by activation of phenylalanine ammonia-lyase (PAL) and β-1,3-glucanase and increased levels of hydrogen peroxide ($H_2O_2$) or superoxide radical ($O_2^-$) generation rate in $GA_3$-treated fruit [16,17].

The effect of the $GA_3OD$ formulation at a range of 500 to 1500 $\mu g \cdot L^{-1}$ was also interesting, showing a reduction in ABS coverage ranging from 56% to 76% compared to the ABS coverage of untreated fruit.

Transformation of the level of infection to the percent of unmarketable fruit after storage showed that 27% of untreated fruit were unmarketable compared to 21% of $GA_3$ sprayed fruit before harvest, and 9% and 13% of fruit when treated with the postharvest dips of $GA_3$ and $GA_3OD$, respectively (Figure 2B). These results indicated again that postharvest $GA_3$ dips were very efficient in the reduction of ABS development.

### 3.3. Effects of Pre and Postharvest Treatment with $GA_3$ on Color Developmnet of Persimmon Fruit

Preharvest treatments of persimmon fruits with 50 $\mu g \cdot L^{-1}$ $GA_3$ in the orchard are sprayed at the color change from green to orange. $GA_3$ treatment usually delays full color development by 7–10 days, after which the fruit becomes orange and is harvested [3]. Untreated fruit however continue the normal color development process, and will show full color development compared to $GA_3$-treated fruit which will show some remnants of green at harvest. In the present experiments, $GA_3$ treatment in the orchard (commercial treatment) were harvested at an average hue value of 69.5 ± 2.3 while untreated fruit were harvested at 67.1 ± 4.0. After 3 months of storage, $GA_3$ sprayed fruits showed a higher hue value of 67 compared to 61 in the untreated fruit (Figure 3). Postharvest treated fruit that were harvest at the same time as the control fruit, showed hue values similar to that of untreated fruit after 3 months of storage. This suggested that postharvest treatment of $GA_3$ at higher concentrations than those used in the field did not affect fruit firmness or ABS area by modulation of fruit maturation, as determined by color development. The improved quality induced by postharvest $GA_3$ treatment compared to the preharvest treatment, suggested that a single postharvest dip treatment may replace commercially-applied preharvest $GA_3$ treatment.

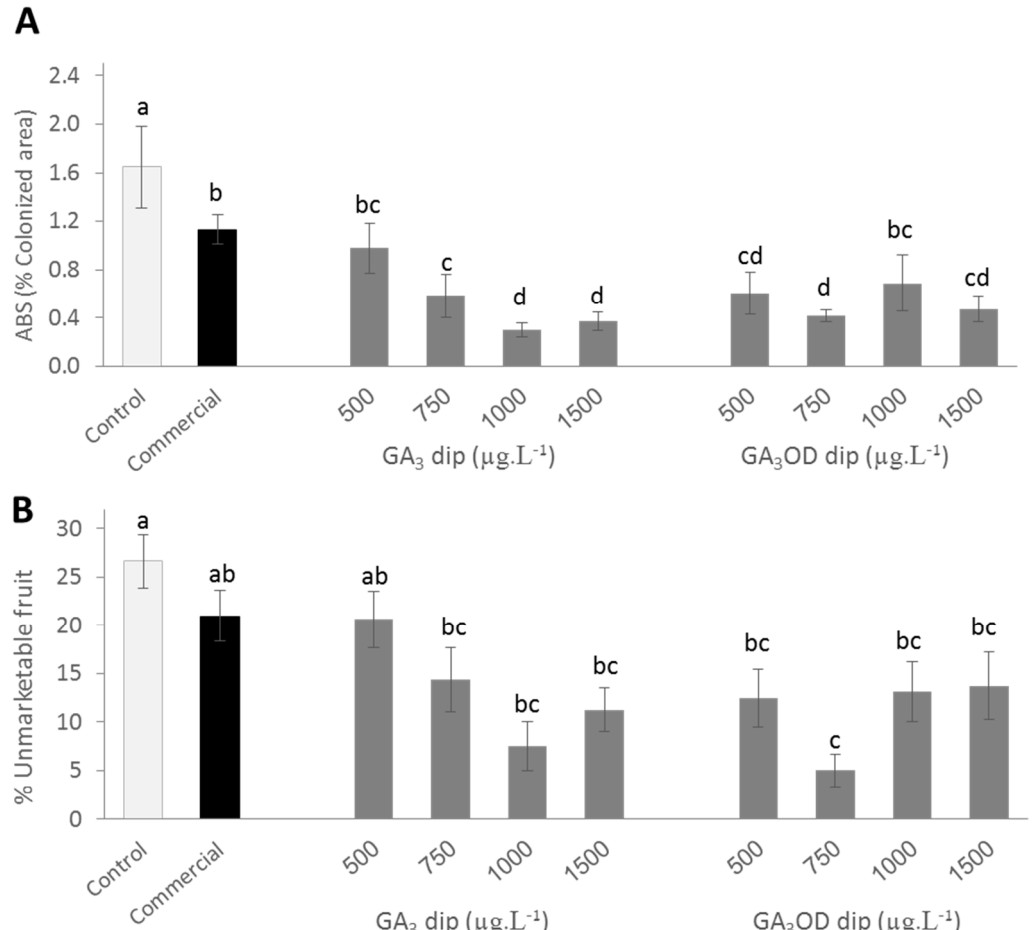

**Figure 2.** Effect of preharvest spray (commercial) and postharvest dip treatments with gibberellin (GA$_3$), using the formulations GA$_3$ and GA$_3$OD, on the levels of Alternaria black spot on persimmon fruit cv (**A**) and percent of unmarketable fruit (**B**). Triumph after 3 months of storage at 0 °C. Average values (±standard error) of six replications with different letters are significantly different at $P \leq 0.05$ according to the Tukey–Kramer Multiple Comparison Test.

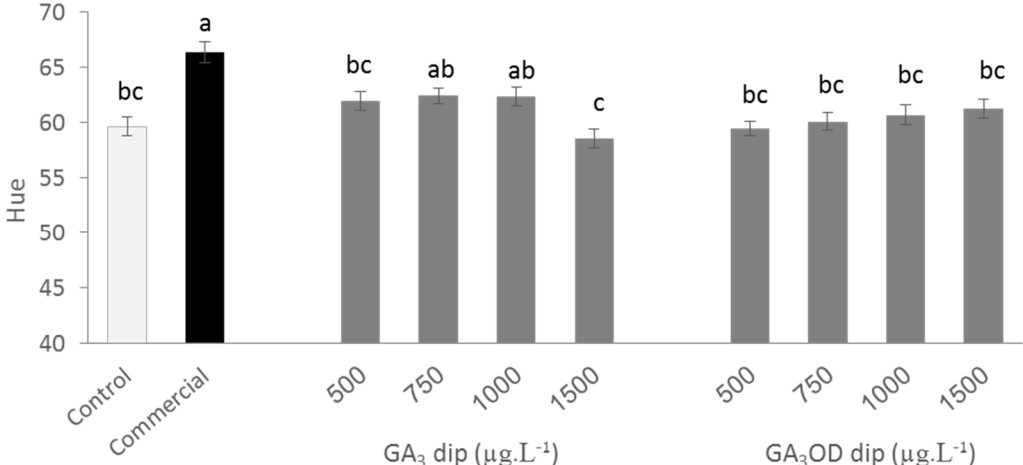

**Figure 3.** Effect of preharvest spray (commercial) and postharvest dip treatments with gibberellin (GA$_3$), using the formulations GA$_3$ and GA$_3$OD, on fruit color of persimmon fruit cv. Triumph after 3 months of storage at 0 °C. Average values (±standard error) of six replications with different letters are significantly different at $P \leq 0.05$ according to the Tukey–Kramer Multiple Comparison Test.

### 3.4. Effect Postharvest GA₃ Treatment under Commercial Conditions in Two Orchards

Comparison of the fruit firmness of the preharvest sprayed to postharvest dip treated fruits from two different orchards showed that both treatments similarly enhanced fruit firmness as well as reduced ABS values after 3 months of storage at 0 °C (Table 1). Firmness values in fruit from the Ein Carmel orchard treated either with a pre- or postharvest treatment showed similar values of ca. 24.7–24.5 N compared to 15.7 N of untreated fruits (Table 1). On fruit from the Bar Ness orchard, firmness values in pre- and postharvest treated fruit ranged from 14.6 to 16.6 N compared to 11.1 N of untreated fruit. There was no difference in fruit firmness between application times.

When the effect of pre- and postharvest treatments were compared for their effect on the ABS index, both treatments significantly reduced ABS severity compared to untreated fruits (Table 1). Pre- and postharvest treatment reduced ABS coverage by 75% compared to untreated fruit in the Bar Ness orchard. In fruit from the Ein Carmel orchard, the postharvest GA₃ dip treatment was even more efficient than the preharvest spray, thereby showing the efficacy of the postharvest treatment under commercial conditions for improving fruit quality during storage.

**Table 1.** Comparison of the effect of GA₃ treatment at pre and postharvest treatments at two different orchards during 2016 and 2017.

| Orchard | Fruit Firmness (N) | | | ABS (%) | | |
| --- | --- | --- | --- | --- | --- | --- |
| | Control | Preharvest Spray 50 µg·L$^{-1}$ | Postharvest Dip 1000 µg·L$^{-1}$ | Control | Preharvest Spray 50 µg·L$^{-1}$ | Postharvest Dip 1000 µg·L$^{-1}$ |
| Ein Carmel (2016) | 15.7 b $^z$ | 24.7 a | 24.5 a | 1.6 a | 1.1 b | 0.3 c |
| Bar Ness (2017) | 11.1 c | 16.6 a | 14.6 ab | 0.8 a | 0.2 b | 0.2 b |

$^z$ Average values of firmness or ABS with differing letters for each orchard indicate significant differences at $P \leq 0.05$ according to the Tukey–Kramer Multiple Comparison Test.

## 4. Conclusions

The present experiments indicated that a simple postharvest GA₃ dip can induce physiological changes that affect fruit softening and reduced ABS development. This indicated that mature fruit that is harvested with an orange color and is treated by postharvest dips still respond to GA₃, thereby enhancing storability. The mechanism of this response is probably not related to delayed maturation as observed in GA₃ [1,3] or GA$_{4+7}$+BA (Superlon) treatments [18] that delayed fruit maturation, but is probably acting as a modulator of host physiological responses that should be further studied. These results highlighted for the first time the capability of persimmon fruits to respond to postharvest treatments with GA₃ enhancing storability of the fruits, similar to the response to the preharvest treatments that have been applied during the last 30 years in Israel.

**Author Contributions:** D.P. contributed in the design of the study, gave the conceptual frame of project, supervised the design of the study, drafted the paper and adjusted the final version of the manuscript; D.M. and O.F. ran the laboratory work, and sampling at harvest and after storage, and processed the data. S.G., A.T., G.A. designed the field experiments and helped with the sampling of the fruit. All the authors have read the final manuscript and approved the submission.

**Funding:** This research was supported by Gadot Agro, Gadot Group, Israel.

**Acknowledgments:** Growers from Bar Ness and Ein Carmel orchards should be grateful for the field experiments carried out in their orchards and the use of their storerooms.

**Conflicts of Interest:** The authors declare no conflict of interest.

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
