# Peer review of "Postharvest Dips of Persimmon Fruit in Gibberellic Acid: An Efficient Treatment to Improve Storability and Reduce Alternaria Black Spot"

_horticulturae, doi:10.3390/horticulturae5010023_

Round 1

Reviewer 1 Report

The authors must attend all the observations made in the manuscript so that it can be accepted.

Author Response

We thank the reviewer for you the significant suggestions to improve the manuscript.

All the suggestions marked in yellow in the version supplied by the reviewer were corrected. On suggestion however in Line 224 of the new version was not followed and is related to the request to transfer the details of the field experiment to the materials and methods was corrected. We believe that given that this is a short manuscript with the results and discussion presented in one paragraph, the details of field experiments should be presented in that location.

Beside that we thank all the suggestions.  

Reviewer 2 Report

The manuscript needs major revision for English language. At places, it is difficult to understand due to poor writing style. Number of writing/formatting errors, some of the references are not properly formatted, (example: p value in line 108; line 121: “Newton”, line 22)  . DO include data from time of harvest on color and firmness so readers can clearly see the effect of storage in untreated control. Since the study was conducted for 3 years, do add any tree physiology or yield information of trees used in pre-harvest treatment (section 2.2). Add error bars to figures. What was the number of replicates for each treatment? What was the experimental unit for untreated control and dip treatment? Were fruit for untreated control and dip treatment harvested from same tree? What was the criteria for unmarketable fruit? Until like 217, there is no clear indication of section 3.4 and 3.5.

Author Response

We thank the reviewer for his corrections for the improvement of the manuscript.

Comments by the reviewer

1.The manuscript needs major revision for English language. At places, it is difficult to understand due to poor writing style.

Response: the ms was re-read and many corrections were included

2. Number of writing/formatting errors, some of the references are not properly formatted, (example: p value in line 108; line 121: “Newton”, line 22)

 Response: the formatting errors were corrected and include in the new version.

3. DO include data from time of harvest on color and firmness so readers can clearly see the effect of storage in untreated control.

Response: the values were added in the text.

4. Since the study was conducted for 3 years, do add any tree physiology or yield information of trees used in pre-harvest treatment (section 2.2).

Response: the data on tree physiology as a result of field GA3 treatment is out of the scope of this manuscript that try to describe the effect of GA3 on the fruit quality after harvest. However, we have quoted some of the results of Winer and Bendik in Line 48 of the new version: "Winer and Bendik (2012) reported that this effect could reach up to 25 and 26.2% decrease in total yield and exported fruits, respectively. "

4. Add error bars to figures.

Response: the bars were added to the figures

5. What was the number of replicates for each treatment?

Response: As described in the line 66: " The effect of Gibberellin treatment on persimmon fruit quality was tested in a random split-block design, spread along four rows of trees, with six replicates"

6. What was the experimental unit for untreated control and dip treatment?

Response: As describe in line 70: "Each replicate comprised 6 trees with three limiting trees separating the replicates." Also in Line 73: " After 3 months of storage, samples of 80 fruit from each bin (80 fruit x 6 bin= 480 fruit) were transferred from the commercial storage room to the Department of Postharvest Science at the ARO, Bet Dagan, Israel, for evaluation."

7. Were fruit for untreated control and dip treatment harvested from same tree?

Response: As described in line 66: " The effect of Gibberellin treatment on persimmon fruit quality was tested in a random split-block design".. the fruits can not be harvest from the same tree.

8. What was the criteria for unmarketable fruit? Until like 217, there is no clear indication of section 3.4 and 3.5.

Response: the criteria for unmaketable is already described in line 95: "The percent of ABS, disease severity, was recorded as the percentage of the surface area exhibiting black spot decay. Fruit was assessed following 3 months of storage at 0 ◦C and (Perez et al., 1994): a fruit was regarded as unmarketable when more than 1% of its surface area was covered by ABS."

We hope that in the present for the ms will be acceptble for publication in Horitculture

Round 2

Reviewer 2 Report

NA